# Exploring Spatial Heterogeneity of Immune Cells in Nasopharyngeal Cancer

**DOI:** 10.3390/cancers15072165

**Published:** 2023-04-05

**Authors:** Aastha Sobti, Christina Sakellariou, Johan S. Nilsson, David Askmyr, Lennart Greiff, Malin Lindstedt

**Affiliations:** 1Department of Immunotechnology, Lund University, 223 81 Lund, Sweden; 2Department of ORL, Head & Neck Surgery, Skåne University Hospital, 221 85 Lund, Sweden; 3Department Clinical Sciences, Lund University, 221 00 Lund, Sweden

**Keywords:** nasopharyngeal cancer, immune phenotypes, spatial omics, digital spatial profiling, biomarker discovery, immune cells

## Abstract

**Simple Summary:**

Nasopharyngeal cancer (NPC) is a malignant tumor of the upper pharynx. In this study, we used multiplex tissue analysis combined with digital spatial profiling to map the immune heterogeneity of NPC. Forty-seven specified regions-of-interest (ROIs) with 49 target proteins were profiled across 30 cases of NPC for quantitative assessment of proteins related to CD45^+^ cells present in NPC tissue. Protein markers associated with B cells, NK cells, macrophages, and regulatory T cells were expressed at higher levels in ‘immune-rich cancer cell islets’ compared to the ‘surrounding stromal leukocyte’ regions. In contrast, biomarkers associated with suppressive populations of myeloid cells and exhausted T cells were overexpressed in ‘surrounding stromal leukocytes’ compared to ‘immune-rich cancer cell islet’ regions of the tumor. Moreover, findings regarding defined cancer phenotypes, such as ‘inflamed’, ‘immune-excluded’, and ‘desert’, were highlighted. In the ‘inflamed’ phenotype, compared with the other two phenotypes, markers associated with B cells, NK cells, macrophages, and myeloid cells were expressed at a higher level. On the other hand, in the ‘immune-excluded’ phenotype, markers associated with suppressive populations of myeloid cells and T cells were expressed at a higher level compared with ‘inflamed’ and ‘desert’ groups, while the ‘desert’ profile had greater levels of granulocyte markers and immune-regulatory markers. The results shed light on the cellular composition of NPCs and may aid in stratifying patients to treatment based on their immune microenvironment.

**Abstract:**

Nasopharyngeal cancer (NPC) is a malignant tumor. In a recent publication, we described the presence and distribution of CD8^+^ T cells in NPC and used the information to identify ‘inflamed’, ‘immune-excluded’, and ‘desert’ immune phenotypes, where ‘inflamed’ and ‘immune-excluded’ NPCs were correlated with CD8 T cell infiltration and survival. Arguably, more detailed and, in particular, spatially resolved data are required for patient stratification and for the identification of new treatment targets. In this study, we investigate the phenotype of CD45^+^ leukocytes in the previously analyzed NPC samples by applying multiplexed tissue analysis to assess the spatial distribution of cell types and to quantify selected biomarkers. A total of 47 specified regions-of-interest (ROIs) were generated based on CD45, CD8, and PanCK morphological staining. Using the GeoMx^®^ Digital Spatial Profiler (DSP), 49 target proteins were digitally quantified from the selected ROIs of a tissue microarray consisting of 30 unique NPC biopsies. Protein targets associated with B cells (CD20), NK cells (CD56), macrophages (CD68), and regulatory T cells (PD-1, FOXP3) were most differentially expressed in CD45^+^ segments within ‘immune-rich cancer cell islet’ regions of the tumor (*cf*. ‘surrounding stromal leukocyte’ regions). In contrast, markers associated with suppressive populations of myeloid cells (CD163, B7-H3, VISTA) and T cells (CD4, LAG3, Tim-3) were expressed at a higher level in CD45^+^ segments in the ‘surrounding stromal leukocyte’ regions (*cf*. ‘immune-rich cancer cell islet’ regions). When comparing the three phenotypes, the ‘inflamed’ profile (*cf.* ‘immune-excluded’ and ‘desert’) exhibited higher expression of markers associated with B cells, NK cells, macrophages, and myeloid cells. Myeloid markers were highly expressed in the ‘immune-excluded’ phenotype. Granulocyte markers and immune-regulatory markers were higher in the ‘desert‘ profile (*cf.* ‘inflamed’ and ‘immune-excluded’). In conclusion, this study describes the spatial heterogeneity of the immune microenvironment in NPC and highlights immune-related biomarkers in immune phenotypes, which may aid in the stratification of patients for therapeutic purposes.

## 1. Introduction

Nasopharyngeal cancer (NPC) is a squamous cell carcinoma of the upper pharynx (i.e., the nasopharynx). It is a rare malignancy in most parts of the world, with an age-standardized incidence rate (ASR) of less than 1 per 100,000 for both genders in most countries and 5.6 per 100,000 in East Asia [1,2]. NPC is strongly associated with the Epstein-Barr virus (EBV), but environmental variables such as alcohol abuse, tobacco-smoking, and ingestion of salted fish may also contribute [3,4]. In a recent review, Luo highlights the ecological characteristics of NPC and advocates that it is due to a multidimensional pathological ecosystem [5]. The clinical staging system based on the tumor–node–metastasis (TNM) classification, describing the extent of disease, is arguably limited in its prognostic value. While the association (or not) between NPC and the Epstein-Barr virus also carries prognostic information and affects treatment decisions, more precise prognostic markers and effective stratification methods are required [6].

Progression of NPC is associated with a distinct tumor microenvironment [7], and diverse features can be exploited for immune and molecular stratification, treatment response prediction, and prognosis [8]. T cells, B cells, mast cells, macrophages, and neutrophils have been reported to be abundant in NPC, and an increase in macrophage, mast cell, and neutrophil infiltration has been linked to tumor growth [9,10,11]. Furthermore, high expression of immune checkpoint (IC) markers such as PD-1, Tim-3, and LAG-3, as assessed by transcriptomics and immunohistochemistry (IHC), has been observed [12,13,14]. Liu *et al.* used multiplex quantitative fluorescence analysis of 17 immune biomarkers in a cohort of 54 NPC biopsies [12]. They showed that high expression of PD-L1 on intratumoral CD163^+^ macrophages was associated with distant metastatic disease [12]. Additionally, expression of CD117 on mast cells and CXCR5 on follicular helper T cells was associated with an adverse prognosis [12]. Utilizing single-cell RNAseq analysis, Chen *et al.* demonstrated that improved survival correlated with signatures of macrophages, plasmacytoid dendritic cells (pDCs), CLEC9A^+^ DCs, natural killer (NK) cells, and plasma cells [13]. In contrast to previous research, which found an increased frequency of CD163^+^ macrophages to be linked with poor survival [10,12], Chen *et al*. found that macrophages based upon single-cell signatures were associated with improved survival without any further sub-classification into subsets [13]. Furthermore, using computational pathology, quantitative analysis of PD-L1, B7-H3, B7-H4, IDO-1, VISTA, ICOS, LAG-3, Tim-3, and OX40 demonstrated high expression of all markers, apart from LAG-3 and Tim-3, in NPC-associated immune cells [14]. However, the expression profiles were heterogenous between patients, and a substantial part of the cohort did not show increased expression of the above-mentioned IC markers [14].

With the advancement of digital spatial technologies, it is now possible to perform cell-specific and spatially resolved analyses of immune biomarker expression levels in patient samples. This may provide more objective perspectives and better prognostic information than traditional IHC-based pathological estimations [15]. Due to the contribution of both cancer and stromal cells to the formation of the TME and infiltration of tumor-associated immune cells, it is imperative to be attentive to both compartmental and cellular aspects. Moreover, disparities in the immune responses among patients with different clinical outcomes must be addressed and highlighted. In the present study, the GeoMx^®^ digital spatial profiler (DSP) was used to investigate differential protein expression in areas of interest (AOIs) from ‘immune-rich cancer cell islets’ and ‘surrounding stromal leukocyte’ regions for 30 treatment-naive biopsies from NPC patients. Differential protein expression analysis was performed to further develop previously identified immune phenotypes: ‘inflamed’, ‘immune-excluded’, and ‘desert’ [16]. Compared with the latter two, the ‘inflamed’ profile had higher expression levels of IC markers associated with T cells, B cells, and NK cells. On the other hand, markers associated with granulocytes and fibroblasts were seen to be increased in the ‘desert’ and ‘immune-excluded’ phenotypes compared with those in the ‘inflamed’ group. Furthermore, the prognostic impact of detected differentially expressed biomarkers was assessed. The findings of this study deepen our understanding of the correlation between protein markers and spatial distribution of immune cell populations, which may help in characterizing and stratifying NPC patients for therapy.

## 2. Materials and Methods

### 2.1. Sample Cohort

Formalin-fixed paraffin-embedded (FFPE) tissue samples from 42 patients diagnosed with NPC between 2001–2015 were collected from the Department of Pathology, Skåne University Hospital, Sweden, after approval by the ethics review authority (2014/117). The samples were morphologically assessed using hematoxylin–eosin and CD8 staining (anti-CD8 Ab, clone C8/144B, Dako/Agilent, Glostrup, Denmark) to identify and mark cancer cell areas and immune-rich regions within the tumors. Tissue microarrays (TMA)were prepared by extracting 1.5 mm wide cylindrical tissue cores from the slides. Of the 42 patients, the quality of the 32 biopsies was considered adequate for analysis. Clinicopathological information associated with the samples is shown in Appendix A.

### 2.2. Digital Spatial Profiling (DSP)

Glass slides containing 5 µm sections of the FFPE TMAs were stained with DNA stain (SYTO™ 13 Green Fluorescent Nucleic Acid Stain, Invitrogen, Waltham, MA, USA), fluorescently labelled antibodies against CD8 (CD8-AF666/Cy5, clone OTI3H6, Origene, Rockville, MD, USA), PanCK (pan-Cytokeratin-AF568/Cy3, clone AE1 + AE3, Novus Biologicals, Littleton, CO, USA), and CD45 (CD45-AF615/Texas Red, clone 2B11 + PD7/26, Novus Biologicals, Eaglewood, CO, USA) (Appendix A and Table 1). In addition, 43 target-specific oligonucleotide-tagged antibodies conjugated with a UV-cleavable linker were added to the TMA slides. These antibodies were from four predesigned panels: ‘Immune cell profiling’, ‘Immunology drug targets’, ‘Immune activation status’, and ‘Immune cell typing’ (NanoString Technologies, Seattle, WA, USA) (Appendix A). In addition, three control housekeeping proteins (GAPDH, Histone H3, and S6) and three negative isotype controls (Ms IgG1, Ms IgG2a, and Rb IgG) were added as part of the protein mixture. The slides were visualized using a NanoString GeoMx^®^ DSP instrument (NanoString Technologies, Seattle, WA, USA). Within each tumor core, two locations were chosen as regions of interest (ROIs): ‘immune-rich cancer cell islets’ and ‘surrounding stromal leukocyte’ regions. Next, areas of illumination (AOIs) were generated based on cell expression of morphological markers (CD45^+^, CD8^+^, and PanCK^+)^ from the regions of interest (ROIs). After UV exposure, the bound reagents in the AOIs were collected into separate wells on a microtiter plate and quantified using the nCounter^®^ flex analysis platform (NanoString Technologies).

### 2.3. Statistical Analysis

Quality control and obtained counts for each protein marker in the AOIs were processed using GeoMx DSP Analysis Suite (NanoString Technologies) [15]. Normalization using housekeeping proteins (GAPDH and Histone H3) and area scaling was conducted to adjust for the discrepancies in the sizes of different AOIs and cells. Further analyses were performed using RStudio (Boston, MA, USA) (R Version 4.2.1, Vienna, Austria). Protein expression levels of CD45, CD8, and PanCK were assessed to validate the segmentation strategy of the AOI collection. Multiple segments from each tumor core were collected, and CD45^+^ segments were further subjected to the linear mixed model, keeping the tumor cores and proteins as the dependent factor to account for variability via the glmmSeq package [17]. Log2-transformed counts were considered for two-group analysis of immune cells between cancer cell islets and the surrounding stromal regions using the Mann–Whitney test with Benjamini, Krieger, and Yekutieli correction. Due to the limited number of samples, notably from patients with EBV-negative NPC, no further analysis was conducted based on EBV presence. Based on our previous classification of samples into three immunophenotypes [16], additional multigroup and two-group assessments were performed using the Kruskal–Wallis and Mann–Whitney tests, respectively. Differentially expressed (DE) proteins were plotted using the Complex Heatmap package [18,19] and EnhancedVolcano package [20]. Survminer and survival packages were used for survival, Cox regression and Kaplan–Meier analyses [21,22]. Cox regression analysis was conducted both as univariate and multivariate analyses. Possible clinical confounding factors, i.e., TNM staging, age, gender, EBV-status, and identified phenotypes, were investigated as predictor variables along with the proteins of the initial multivariate analysis. Additional survival analysis was also performed using the head and neck squamous carcinoma (HNSC) dataset from the Cancer Genome Atlas (TCGA) consisting of whole tissue RNA-seq data from 528 HNSC samples [23]. The expression values in the first and fourth quartiles were compared using Kaplan–Meier analysis.

### 2.4. Quantification of CD45 and CD8 Expression by Immunofluorescence Microscopy

Based on CD45 and CD8 staining, cell quantification was performed within the ‘immune-rich cancer cell islet regions’ and ‘surrounding stromal leukocytes’ using the Stardist machine-learning algorithm in the QuPath software (version 0.3.2) [24,25]. Quantification was conducted by identifying all 30 TMA cores using the TME dearrayer in QuPath. The total number of cells was determined using the StarDist algorithm. Spatial phenotyping was based on QuPath’s machine-learning protocol and was further carried out by initially selecting small segments in training images accounting for 30 cells per cell phenotype in five different TMA sections. This classification was applied to the whole TMA, identifying CD45^+^ and CD8^+^ immune cells as well as PanCK^+^ tumor regions. In addition, this information was translated into cancer cell islets which were PanCK^+^, with or without CD45^+^ cells, and surrounding stromal regions consisting of all CD45^+^ cells, inclusive of CD8^+^ cells. Moreover, the ratio of CD45 and CD8 in cancer cell islets and stromal regions over the total cell count was evaluated separately. This was further used to determine the ratio of CD8 and CD45 over the total cell count in the two previously described segments.

### 2.5. Transcriptomics Analysis

Publicly available transcriptomic datasets were used for various validations. Two single-cell RNA sequencing datasets, consisting of 15 and 10 treatment-naive NPC samples with a total of 47,866 cells and 4729 myeloid cells were downloaded from Gene Expression Omnibus (GEO, GSE150430) and www.panmyeloid.cancer-pku.cn (accessed on 25 August 2022), respectively [13,26]. To enable biomarker comparisons from both datasets, single-cell and myeloid populations were defined according to their established gene signatures [13,26]. These predefined populations aided in correlating the gene distribution of our immune cells’ associated protein markers of interest (CD27, CD4, CD11c, IDO1, and Fibronectin) within the predefined single-cell populations in [13,26].

## 3. Results

### 3.1. Intra-Tumor Spatial Profile of NPC for Selected CD45^+^ Regions Show an Inhibitory Profile in Surrounding Stromal Regions

Of the 32 samples analyzed, 2 were excluded from the analysis due to detachment from the slide or lack of cancer cells for identification of regions. The clinical characteristics of the remaining 30 patients are presented in Appendix A. Using the CD45 staining, immune cells from different regions were selected from the cores of the NPC samples. A total of 47 AOIs from CD45^+^ areas were obtained and further divided into 18 ‘immune-rich cancer cell islets’ and 29 ‘surrounding stromal leukocyte’ regions (Figure 1a). After accounting for biological variances within each tumor using normalization and the linear mixed-effects model, principal component analysis revealed a clear distinction between the immune cell profiles in ‘immune-rich cancer cell islets’ and ‘surrounding stromal leukocyte’ regions (Figure 1b and Appendix A). Twenty DE proteins were observed between these two segments (Figure 1c). Fibronectin, CD4, CD14, CD163, B7-H3, LAG3, Tim-3, PD-L1, VISTA, CD25, CD45RO, and CD44 were significantly highly expressed in ‘surrounding stromal leukocytes‘ regions, whereas the expression levels of GZMB, CD20, CD56, CD68, Ki-67, PD-1, FOXP3, and CD45 were higher in ‘immune-rich cancer cell islet’ regions. B7-H3 and CD14 had a positive correlation (r = 0.71, *p*-value = 2.94 × 10^−8^), as did PD-1 with CD20 and CD56 (r = 0.79, *p*-value = 2.90 × 10^−11^ and r = 0.7, *p*-value = 4.66 × 10^−8^, respectively) (Figure 1d). CD68 was negatively correlated with CD4 (r = −0.74, *p*-value = 3.05 × 10^−9^) (Figure 1d).

### 3.2. Inter-Tumor Protein Expression Associated with Three Different Phenotypes

In a previous study, we used quantitative whole-tissue CD8^+^ T cell immunohistochemistry staining to classify the FFPE NPC tissues into ‘inflamed’, ‘immune-excluded’, and ‘desert’ immune phenotypes [16]. In this study, using the same cohort, we verified the phenotypes with CD45 and CD8 immunofluorescence microscopy for the samples included in the DSP analysis (Figure 2a). TMAs were further quantified with regard to the total number of CD45 and CD8 nuclei for each tumor core. These estimates were subdivided into ‘immune-rich cancer cell islets’ and ‘surrounding stromal leukocyte’ regions (Appendix A). Out of 30 biopsies, cancer cell islets revealed CD45^+^ infiltration of 40% in 18 ‘inflamed’ samples, between 0–25% in 9 ‘immune-excluded’ samples, and 0–10% in 3 with ‘desert’ profiles (Appendix A). Only one sample deviated from our former classification. This biopsy was still termed as ‘inflamed’ despite having a total of 25% CD45^+^ infiltration in cancer cell islets compared with 40% in the other 17 inflamed samples (Appendix A). This was based upon whole slide analysis conducted prior to NPC core selection for the TMA as well as CD8^+^ infiltration on whole slide section conducted in our previous analysis [16]. A clear division of CD45^+^ cells in surrounding stromal leukocytes was observed to be less than 10% in the ‘desert’ phenotype compared to the ‘immune-excluded’ and ‘inflamed’ phenotypes (Appendix A).

The CD45^+^ segments of ‘immune-excluded’ and ‘desert’ NPC phenotypes were from ‘surrounding stromal leukocyte’ regions, while CD45^+^ segments of the ‘inflamed’ NPC phenotype were equally distributed between ‘immune-rich cancer cell islets’ and ‘surrounding stromal leukocyte’ regions (Appendix A). Due to the lack of infiltrating immune cells in ‘immune-rich cancer cell islets’ in ‘immune-excluded’ and ‘desert’ tumors, a comparative examination of DE proteins could only be performed using the stromal segments, as expected. Unbiased PCA showed significant differences between the three phenotypes on the 1st component (23.9%) (Figure 2b). Multigroup analysis revealed 13 DE protein markers in the three phenotypes (Kruskal–Wallis test, *p* < 0.05) (Figure 2c). Two-group analysis revealed a higher expression of immune markers such as CD20, CD3, CD56 FOXP3, and Ki-67 as well as ICs such as PD-1, ICOS, and IDO1 in AOIs of ‘inflamed’ NPC *cf*. ‘immune-excluded’ and increased expression of proteins associated with antigen presentation HLA-DR, CD40, and CD80 *cf.* ‘desert’ NPC (Mann–Whitney test, *p*-value < 0.05) (Figure 2d,f). The ‘immune-excluded’ *cf.* ‘inflamed’ showed significantly higher expression of Fibronectin, B7-H3, CD25, CD163, CD44, and CD34 as well as of ICs such as Tim-3, OX40L, and PD-L2 and higher expression of HLA-DR and the IC PD-L2 *cf.* ‘desert’ phenotype (Figure 2d,e). In contrast, the ‘desert’ *cf.* ‘inflamed’ or ‘immune-excluded’ profiles had higher expression of the protein markers FOXP3, Ki-67, CD66b, ARG1, and 4-1BB along with higher expression of additional proteins such as STING, CTLA-4, and SMA upon comparison with the ‘inflamed’ phenotype (Figure 2e,f).

Multigroup analysis in stromal segments revealed differences between tumor stages based on the TNM staging system (UICC’s TNM classification system, 7th version) (Kruskal–Wallis test, *p*-value < 0.05) (Appendix A and Appendix A). Stage I was excluded from this analysis due to the cohort’s small size of only two samples. Other staging groups were made up of at least three NPC samples (like in stage II). Regulatory markers, such as FOXP3 and Tim-3, were higher in stage II NPC than in other stages. Markers such as CD11c and CD40, which are associated with antigen-presenting cells, were enhanced in stage III NPC, and the expression of immune-checkpoint markers PD-L1 and IDO1, and other proteins such as GZMB and SMA, were highest in stage IVA compared with the rest. Immune-related markers CD27 and CD127 and a marker associated with fibroblasts, i.e., Fibronectin, were highly expressed in stage IVC *cf.* other stages. Furthermore, the significantly DE proteins based upon stromal segment expression of the three defined phenotypes (Figure 2c) were merged with the significant DE proteins from the TNM staging (Appendix A) to identify positive or negative expression across clinical stages and phenotypes (Appendix A). Positive or negative expression was calculated based on each group’s average expression of markers. The markers in the 1st quadrant were termed negative and those in the 4th quadrant positive, while the expression of markers falling in the 2nd and 3rd quadrants were termed null or blank. This study helped us to correlate established TNM staging to our specified phenotypes at the protein expression level in the ‘surrounding stromal leukocyte’ regions. Amongst the DE markers, positive IDO1 and ICOS expression was associated with ‘inflamed’ and ‘desert’ profiles, whereas Fibronectin and B7-H3 expression was associated with ‘immune-excluded’ and ‘desert’ profiles. CD11c, CD34, and OX40L were only expressed in the ‘immune-excluded’ phenotype, and markers such as ARG1 and CD66 positively associated with the ‘desert’ phenotype. CD20, CD40, PD-1, PD-L1, and CD27 were expressed in the ‘inflamed’ phenotype.

### 3.3. CD11c and IDO1 Proteins Are Associated with Favorable Overall Survival

Cox regression analysis was used to predict overall survival using the total expression of 43 proteins from all regions of the 30 NPC samples. CD4, Fibronectin, CD27, CD11c, and IDO1 were the protein markers from ‘immune-rich cancer cell islets’ and ‘surrounding stromal leukocyte’ regions linked with a significant hazard ratio and overall survival (Figure 3a). Positive hazard ratios associated with poor survival were observed for the markers CD4, Fibronectin, and CD27 (*p*-value < 0.05). On the other hand, CD11c and IDO1 were associated with negative hazard ratios and better overall survival (*p*-value < 0.05) (Figure 3a). These observations were made in univariate as well as multivariate analyses (Appendix A). Moreover, a Kaplan–Meier-based survival analysis based upon dividing the expression of the markers into higher (>50%) and lower (<50%) expression groups was conducted. This indicated an improved overall survival with higher CD11c and IDO1 expression and lower Fibronectin and CD4 expression (*p*-value < 0.05) (Figure 3b). These observations held true even when analyzed with respect to overall survival (normalized over 7 years) and disease-free survival (overall as well as normalized over 7 years) (Appendix A). Although the enhanced presence of CD11c was observed in ‘immune excluded’ *cf.* ‘inflamed’ and ‘desert’ NPCs, no significant survival differences were observed between the three phenotypes with combined marker expression (based on *p*-value < 0.05). Furthermore, we examined the HNSC dataset from TCGA [23], which includes multiple subtypes of HNSC but no NPC samples (Appendix A). IDO1 and CD4 expression levels were shown to be significant when comparing quartile expression levels within the HNSC dataset. Of the assessed markers, only IDO1 expression correlated with better survival in both TCGA HNSC and our in-house dataset.

To assess immune cells that expressed these biomarkers, we next investigated the expression patterns of CD4, CD27, CD11c/ITGAX, IDO1, and Fibronectin/FN1 across cell types by using publicly accessible single-cell mRNA datasets from NPC samples (Appendix A) [13,26]. A comparative analysis using the twelve immune cell populations in NPC biopsies defined by Chen *et al.,* [13] indicated that T cell- and B cell-associated populations expressed CD27 (Appendix A). CD4 mRNA expression was not only detected in the CD4^+^ T cell population but also in monocytes, DCs, and macrophages. CD11c/ITGAX and IDO1 were expressed mainly by myeloid cells, and Fibronectin by fibroblasts and monocytes/macrophages. By utilizing a dataset from Cheng *et al.,* [26], focusing on the expression patterns of nine predefined subpopulations, including myeloid cells and plasmacytoid DCs in NPC, it was evident that CD11c/ITGAX was broadly expressed by all myeloid populations as well as NK cells (Appendix A). CD4 expression was observed in all defined populations, whereas Fibronectin was observed only in ISG15^+^ and SPP1^+^ macrophages. IDO1 was observed in conventional DCs (cDC1), activated (LAMP3^+^) DCs, and cDC2s as well as in ISG15^+^ and SPP1^+^ macrophages. The expression of CD27 was not detected in any of the myeloid populations in this dataset.

## 4. Discussion

In this study, we examined immunological aspects of NPC in a spatial context by analyzing the expression of a panel of proteins in ‘immune-rich cancer cell islets’ and in ‘surrounding stromal leukocyte’ regions. Furthermore, we did so in subsets of the material defined by the presence and distribution of CD45^+^ cells, i.e., ‘inflamed’ (CD45^+^ cells throughout the tissue), ‘immune-excluded’ (CD45^+^ cells throughout the surrounding stroma but not in cancer cell islets), and ‘desert’ (no/very few CD45^+^ cells) phenotypes of NPC. Importantly, all protein analyses were normalized against the presence of immune cells, allowing for comparisons irrespective of the cell presence. Intra-tumor analysis revealed an immune-inhibitory profile in ‘surrounding stromal leukocyte’ regions of NPC, with higher expression of immunosuppressive markers and lower expression of immune-related markers. Furthermore, the different tumor stages and immune phenotypes were associated with various DE protein markers. Analysis of the 43 proteins from all selected regions within the NPC samples, independent of their intra-tumor spatial profile, revealed that CD4, Fibronectin, CD27, CD11c, and IDO1 were common protein markers linked with overall survival. Notably, CD11c and IDO1 were associated with better overall survival, while CD4, Fibronectin, and CD27 were associated with poor survival. Lastly, single-cell mRNA datasets from NPC samples indicated that CD4 mRNA was expressed by monocytes/macrophages, DCs, and T cells, Fibronectin/FN1 was expressed by fibroblasts and monocytes/macrophages, while CD11c/ITGAX and IDO1 were expressed mainly by myeloid cells. The results of this study provide evidence of a spatially heterogeneous distribution of leukocytes in NPC and how this is associated with clinically relevant outcomes, which may be further used to categorize pre-treatment NPC patients to predict prognosis according to how proteins are expressed.

This study highlights distinct immunological profiles of ‘immune-rich cancer cell islets’ and ‘surrounding stromal leukocyte’ regions in NPC, where ‘immune-rich cancer cell islets’ were only found in the ‘inflamed’ phenotype and ‘surrounding stromal leukocyte’ regions were from all the three phenotypes. Higher expression of proteins associated with T cells, B cells, NK cells, and macrophages (such as GZMB, FOXP3, PD-1, CD20, CD56, and CD68) were observed in the ‘immune-rich cancer cell islets’ *cf.* ‘surrounding stromal leukocyte’ regions. Previous studies, with few exceptions, have primarily focused on tumor-infiltrating lymphocytes, without distinguishing between the cancer-cell islet regions and surrounding stromal leukocyte regions [8,27,28,29,30]. In our study, we identified the expression of GZMB, CD56, and the regulatory T cell (T_regs_) marker FOXP3 within ‘immune-rich cancer cell islet’ regions. This could imply an increased presence of not only effector T cells and NK cells but also of T_regs_ in close approximation with cancer cells and the suppression of anti-cancer effector activities mediated by other infiltrating leukocytes. The increased presence of FOXP3 T_regs_ and CD68 macrophages have been linked to poor survival and tumor growth in NPC via NF-κB pathway alterations [28,31]. Single-cell studies have shown a correlation between T_regs_ and PD-1 as well as PD-1^+^CD8^+^ T cells in NPC tissue, implying that T_regs_ and exhausted CD8^+^ T cells collectively contribute to an immune-suppressive TME [32,33]. Further, we observed an increased expression of CD20 in ‘immune-rich cancer cell islet’ regions associated with B cells. The CXCL13–CXCR5 B-cell associated axis has been explored by single-cell analysis, where they demonstrate improved survival associated with CXCL13^+^ B cell infiltration [34]. Therefore, further spatial analysis utilizing multiplex protein markers, such as CXCL13 and CXCR5 for CD20^+^ B cells subsets, could potentially be valuable for clinical applications. While we detected higher levels of markers associated with effector responses, such as CD20, GZMB, and CD56 in ‘immune-rich cancer cell islet’ regions, additional expression of markers such as PD-1 and FOXP3 in the same regions may reflect mechanisms associated with tumor growth in NPC.

Higher levels of CD163, CD14, Fibronectin, B7-H3, LAG3, Tim-3, PD-L1, VISTA, CD25, and CD44 associated with fibroblasts, myeloid-derived cells, T cells, and NK cells were found in the ‘surrounding stromal leukocytes’ *cf.* ‘immune-rich cancer cell islet’ regions. A study by Zhang *et al*. using hematoxylin and eosin staining to calculate tumor-infiltrating lymphocytes demonstrated that higher lymphocyte levels in the surrounding stroma, but not in cancer cell islet regions, was associated increased overall survival [35]. Although we did not observe higher expression of markers associated with effector cells in the ‘surrounding stromal leukocyte’ regions *cf.* ‘immune-rich cancer cell islet’ regions, we did observe higher expression of IC markers such as LAG3, Tim-3 (HAVCR2), VISTA, and PD-L1 in the surrounding stroma. Single-cell-sequencing-based linear modelling systems and correlations have demonstrated that HAVCR2 and LAG3 are significantly associated with exhausted and T_regs_ in NPC TME *cf.* healthy controls [34]. Wang *et al*. used digital pathology to reveal that ICs such as PD-L1, B7-H3, B7-H4, and IDO-1 were expressed by both tumor cells and associated immune cells in NPC. On the other hand, the IC markers LAG3, VISTA, TIM3, ICOS, and OX40 were only expressed by immune cells in the surrounding stroma [36]. This may be relevant because the spatial location of these markers in the stroma, as also seen in our analysis, makes them a potential therapeutic target in NPC. Additionally, as reviewed by Huang *et al*., VISTA is expressed both on antigen-presenting cells and T cells, but the functional role of VISTA in the TME is still unclear [37]. The presence of IC markers in the stroma surrounding cancer cell islets suggests that the stromal–cancer cell junction comprises an inhibitory barrier. Accordingly, immunotherapy may be an effective treatment option to regulate immune responses in the surrounding stroma prior to chemo/radiotherapy.

Our data recognize a clear division among samples with regard to CD45 infiltration of cancer cell islets and the surrounding stroma, based on the total population of CD45 cells in these regions. Furthermore, the multigroup analysis revealed a handful of DE markers to be associated with each phenotype. In this study, the ‘inflamed’ phenotype *cf.* other phenotypes had a higher expression of markers such as PD-1 and ICOS. Interestingly, we observed the expression of IDO1 in both ‘inflamed’ and ‘desert’ phenotypes in ‘surrounding stromal leukocyte’ regions. IDO1 mRNA expression has previously been identified in cDC2 and activated DCs, with an increase in IDO1 observed during the evolution from CXCL9^+^ cDC2 to LAMP3^+^cDC, suggesting a link between IDO1 and DC maturation [26]. IDO1 has also been observed to be secreted by some fibroblasts, regulating T cell immunity in the NPC TME [13]. In our findings, the higher expression of IDO1 on opposite immune profiles in ‘inflamed’ and ‘desert’ phenotypes might be explained by the expression of IDO1 on different cell types. Furthermore, in our study, IDO1 and PD-L1 expression increased in stage IVA NPC but decreased in stages IVB and IVC. These results might be attributed to a lack of IC biomarkers in the latter stages, even before induction of treatment, resulting in a lower response to IC treatment such as IDO1 inhibitors and PD-1/PD-L1 blockers in recurrent advanced NPC [38,39,40]. Furthermore, in the TCGA HNSC dataset [23], higher IDO1 expression was associated with better survival. The available HNSC samples in TCGA are a mixed cohort of malignancies, including both immune-rich (e.g., tonsils) malignancies and those with low immune activity (e.g., tongue), however data from NPC samples are lacking. As a result, this suggests that IDO1 might serve as a prognostic marker in HNSC. Higher expression of CD11c, a marker expressed by DCs, macrophages, and monocytes, was associated with improved survival and was seen in ‘immune-excluded’ *cf.* ‘inflamed’ and ‘desert’ phenotypes as well as in stage III NPC. These findings suggest that the expression of CD11c and IDO1 may be associated with a variety of immune cell compositions and phenotypes in NPC. Further studies should focus on elucidating the role of these proteins in the TME of NPC.

In the present study, higher expression levels of B7-H3 as well as Fibronectin were observed in ‘surrounding stromal leukocyte’ regions of the ‘immune-excluded’ phenotype *cf.* the ‘inflamed’ phenotype. These two markers have been suggested to contribute to the inhibition of T cell function and to enhance tumor cell motility [41,42]. Furthermore, transcriptomic expression of Fibronectin in NPC tissue as well as in peripheral blood has been associated with advanced clinical stages and poor survival [43,44]. Similarly, in the present study, higher expression of Fibronectin in NPC correlated with poor survival. Additional analyses should also be conducted to examine the potential differences in overall survival between tumor phenotypes in larger cohorts harboring different levels of these biomarkers. Apart from the above markers, the granulocyte marker ARG1 was highly expressed in ‘desert’ *cf.* ‘inflamed’ and ‘immune-excluded’ NPC (in this study). Using multiplex staining, ARG1 expression has been associated with tumor-associated macrophages in HPV^-^ head and neck cancers and poor prognosis [45]. According to observations in a mouse model, the ARG1 pathway promotes proliferation and tissue repair while supporting tumor growth [46].

Taken together, our results provide valuable insight into the immune system’s role in NPC. In particular, the differences observed in immune cell profiles provide evidence that tumors are composed of multiple TMEs with distinct profiles. This suggests that further research should focus on understanding how these TMEs interact to affect the progression of NPC. Additionally, correlations between different DE proteins may provide clues as to how these proteins interact to affect the disease outcome.

## 5. Conclusions

In conclusion, the present study considers the spatial heterogeneity of NPC and demonstrates that IC markers between ‘immune cells in cancer cell islets’ and ‘surrounding stromal leukocyte’ regions vary.The study highlights immunological biomarkers that may be useful as prognostic tools as well as therapeutic targets. It provides novel insights into the spatially heterogeneous distribution of leukocytes in NPC, its association with distinct immune phenotypes, and how this may impact overall survival. One limitation of the study is the small sample size, and our findings in this pilot should be validated in larger cohort of samples. However, it highlights the importance of understanding the spatial profiles of leukocytes to identify prognostic biomarkers predictive of clinical outcomes. Given the role of leukocytes in modulating the TME and their contribution to clinicopathological parameters, a better understanding of spatial profiles may provide new avenues for improved diagnostics and for personalized stratification to treatment in NPC.

## Figures and Tables

**Figure 1 cancers-15-02165-f001:**
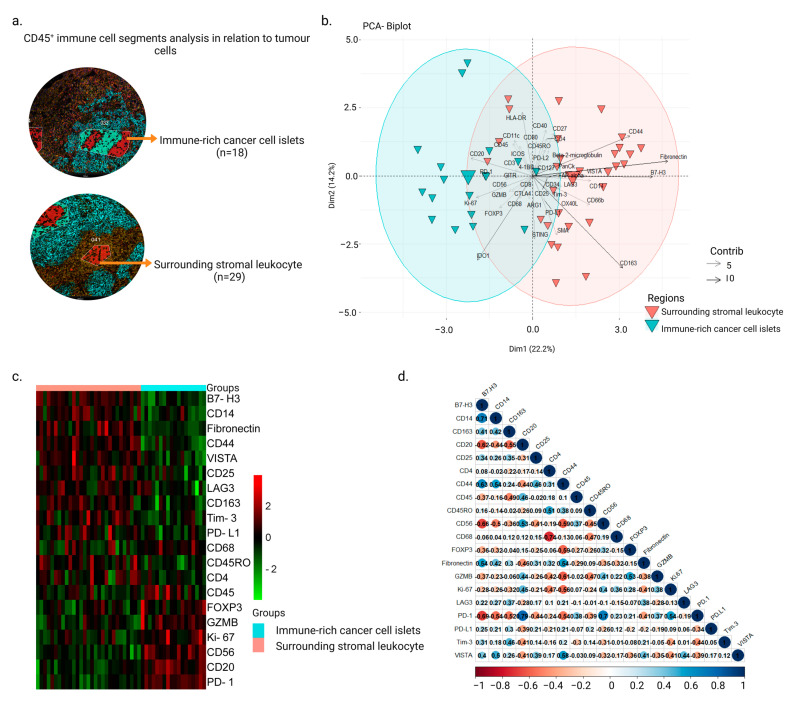
Differential expression profiles in CD45^+^ segments from ‘immune-rich cancer cell islets’ and ‘surrounding stromal leukocyte’ regions in NPC. (**a**) AOIs were generated based on cell expression of the morphological markers CD45 (red), PanCK (cyan), and CD8 (magenta) from the ROIs, and the collected oligonucleotides were quantified using the GeoMx^®^ DSP system. (**b**) Biplot principal component analysis showing ROI segments (AOIs) as well as protein markers on the first and second components from ‘immune-rich cancer cell islets’ and ‘surrounding stromal leukocyte’ regions. (**c**) Heatmap demonstrating DE proteins between ‘immune-rich cancer cell islets’ and ‘surrounding stromal leukocyte’ regions (Mann–Whitney test, *p*-value < 0.05). (**d**) Twenty DE proteins’ correlation map (Pearson’s correlation, significance taken at |r| > 0.7, *p*-value < 0.05).

**Figure 2 cancers-15-02165-f002:**
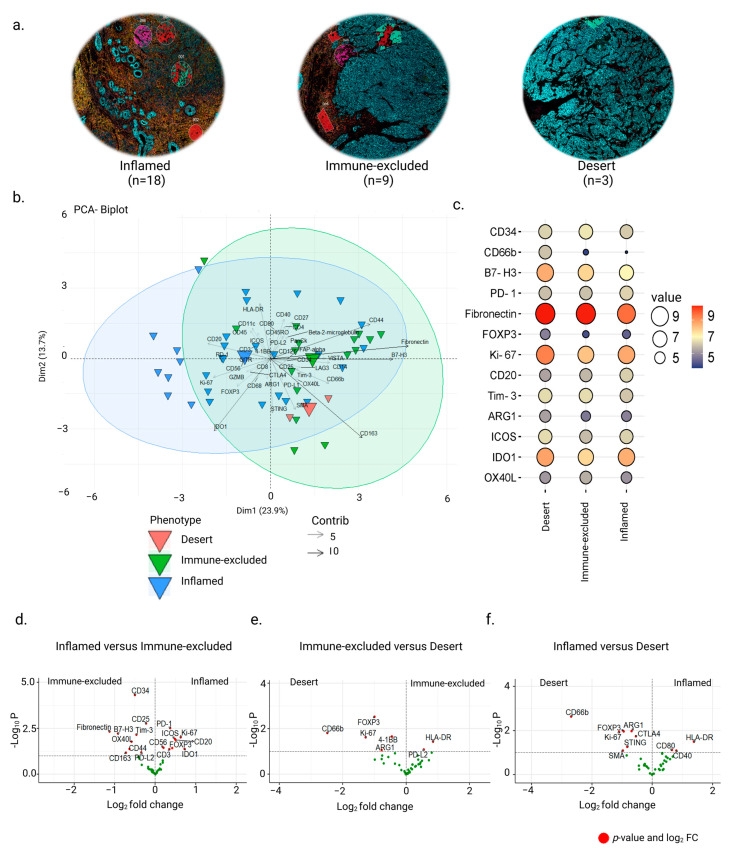
Protein expression in ‘surrounding stromal leukocyte‘ regions associated with immune phenotypes of NPC. (**a**) Representative biopsy sections from the TMA demonstrating the three phenotypes: ‘inflamed’, ‘immune excluded’, and ‘desert’. (**b**) Principal component analysis showing the distribution of the AOIs associated with the three phenotypes. (**c**) Balloon plot showing the top DE proteins between the phenotypes (Kruskal–Wallis test, *p*-value < 0.05). (**d**–**f**) Volcano plots showing differences between the ’surrounding stromal leukocyte’ regions of ‘inflamed’, ‘immune-excluded’, and ‘desert’ phenotypes (Mann–Whitney test, *p*-value < 0.05).

**Figure 3 cancers-15-02165-f003:**
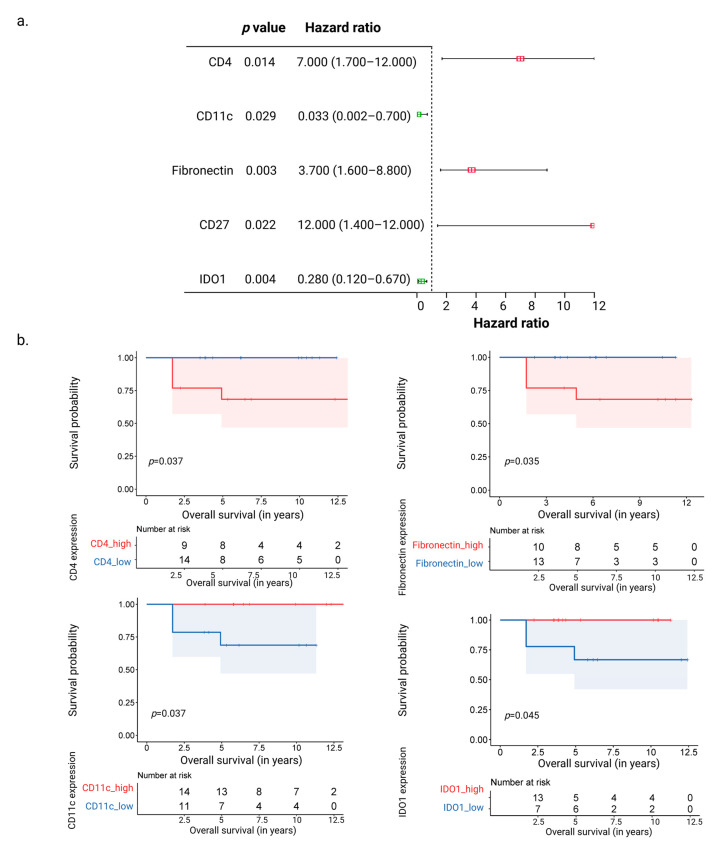
Correlation of the most significant genes between groups and survival. (**a**) Forest plot showing the proteins associated positively (CD4, CD27, and Fibronectin) and negatively (CD11c and IDO1) with hazard ratios associated with overall survival, as seen after Cox regression analysis. (**b**) Kaplan–Meier plots showing the proteins (CD4, Fibronectin, CD11c, and IDO1) associated with the overall survival probability.

**Table 1 cancers-15-02165-t001:** Antibody panel employed for cell identification.

Antibody	Clone	Supplier
CD8	C8/144B	Dako/Agilent
CD8-AF666/Cy5	OTI3H6	Origene
CD45-AF615/Texas Red	2B11 + PD7/26	Novus Biologicals
Pan-Cytokeratin-AF568/Cy3	AE1 + AE3	Novus Biologicals
SYTO™ 13		Invitrogen

## Data Availability

Raw data files from the experiment will be available from the corresponding author on direct request.

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
