# Peer review of "Exploring Spatial Heterogeneity of Immune Cells in Nasopharyngeal Cancer"

_cancers, 2023, doi:10.3390/cancers15072165_

Round 1

Reviewer 1 Report

In this study, GeoMx® Digital Spatial Profiler (DSP) was used to analyse 49 targets proteins in a total of 47 specified regions-of-interest (ROIs) across 30 NPC biopsies. First, the differentially expressed protein targets in CD45+ segments were identified between the 'immune-rich cancer cell islets' regions and ‘surrounding stromal leukocytes' regions of the tumor. Second, the authors had divided the NPC cases into three phenotypes including 'inflamed', 'immune-excluded', and 'desert'. The differentially expressed proteins in the stromal regions were analysed and compared. Third, the 43 proteins from all regions of the NPC samples were analysed for the predictive potential of the overall survival.

This study has shown insightful information to spatially heterogenous distribution of leukocytes in NPC and their clinicopathological values. All these contribute to a big picture of how the TMEs interact to modulate the development and progression of NPC.

However, non-major revisions are needed to clarify some details in the analyses.

Question #1:

In Figure 1, a total of 47 ROIs from CD45+ areas were obtained. 18 of them are attributed as ‘immune-rich cancer cell islets’, while 29 of them are attributed as ‘surrounding stromal leukocytes’ regions. Besides, they are collected from the 18 cases of ‘inflamed cancer cases’, 9 cases of ‘immune-excluded cases’ and 3 cases of ‘desert cases’.

Are all the ‘immune-rich cancer islets ROIs’ taken from the ‘inflammed cancer cases’?

A supplementary table is needed to list which cancer cases contribute to how many ‘immune-rich cancer cell islets’ and ‘surrounding stromal leukocytes’ regions.

I’m wondering how many pair-wise ‘immune-rich cancer cell islets’ and ‘surrounding stromal leukocytes’ ROIs within a NPC case were taken. It may be informative to compare the differentially expressed proteins between these pair-wise ROIs.

In the same tumor, the mutational burden and EBV influence to the TME are the same. These pair-wise analyses can provide additional information about what types of cells can infiltrate into the tumor islets while which types of cells are mostly excluded to the surrounding stroma.

Question #2:

In the paragraphs of 3.2, (Page 6, line 230 – 235):

“The CD45+ segments in 'immune-excluded' as well as all CD45+ segments in 'desert' NPC phenotypes, were from surrounding stroma, whereas CD45+ segments were equally distributed between 'immune-rich cancer cell islets' and 'surrounding stromal leukocytes' in 'inflamed' NPC. As 'immune-excluded' and 'desert' tumors lacked infiltrating immune cells in the cancer cell islets, a comparative analysis of DE proteins could, as expected, only be performed using the stromal segments.”

The message from the above sentences is unclear…..

It means only 29 ROIs of ‘surrounding stromal leukocytes’ were analysed in this section? But there are 30 patient cases. Which NPC case did not contribute to the ROI? If this is from the ‘Desert’ cases, the n number of this phenotype will be 2.

As in Question #1, a supplementary table listing which cancer cases contribute to how many ‘immune-rich cancer cell islets’ and ‘surrounding stromal leukocytes’ regions can clarify the situation.

Question #3:

In page 5, line 198: “the expression levels of GZMB, CD20, CD56, CD68, Ki-67, PD-1, FOXP3, 198 and CD45 were higher in 'immune-rich cancer cell islets' regions.

But in Fig 1c, CD68 is grouped with markers that were significantly highly expressed in ‘surrounding stromal’ regions.

Some explanations may be needed to clarify the presentation of the data.

Author Response

Dear reviewer,

Kind regards,

Authors

Reviewer 2 Report

An interesting study to examine the spatial heterogeneity of the immune microenvironment in NPC and highlights immune-related biomarkers in immune phenotypes by using the GeoMx® Digital Spatial Profiler (DSP). Several points should be noted as below.

1) A inspiring paper has recently proposed NPC ecology theory, and cancer as a multidimensional spatiotemporal unity of ecology and evolution pathological ecosystem. In addition, “spatiotemporal tumoriecology” also is discussed in this paper. More details should be checked in (Nasopharyngeal carcinoma ecology theory: cancer as multidimensional spatiotemporal unity of ecology and evolution pathological ecosystem. Theranostics 2023; 13(5):1607-1631. doi:10.7150/thno.82690. https://www.thno.org/v13p1607.htm). Such views should be updated in the Introduction section.

2) What is the pathological classifications of these NPC patients. Any protein markers in different pathological classification groups? The number cases of NPC analyzed for the survival is limited, it should be declared.  

3) Does EBV infection status contribute to different protein markers in these 30 cases? And how about the correlation with protein markers and EBV-related proteins (e.g. LMP1, LMP2A).

4) In Figure 3b “Kaplan Meier plots showing the proteins (CD4, Fibronectin, CD11c, and IDO1) associated with the overall survival probability.”, the total cases were 30 at the first following up? It should be clear.  

Author Response

Dear reviewer,

Kind regards,

Authors

Reviewer 3 Report

The paper "Exploring spatial heterogeneity of immune cells in nasopharyngeal cancer" by Sobti and colleagues describes the profiling of 43 proteins in various immune cells from FFPE slides from previously analyzed nasopharingeal cancer (NPC) samples. The study is interesting but it could be made more robust and convincing. Below, my comments and concerns.

- While I understand the special issue this paper is intended to is about EBV, the virus itself has no presence in this study. I would suggest leaving the statement in the introduction line 58 to establish a connection between NPC and EBV, but removing the EBV reference from the summary, as it misleads the reader into thinking this study is about EBV-related NPC.

- The claim of association between survival and immune population phenotype is strong, but I believe it is not fully proven. The analysis shown in figure 3b is not convincing (also given the fact that, having calculated three p-values in the 0.03-0.04 range, an even mild FDR correction would make all of them not-significant). The authors must use the markers predicted by their study (CD4, Fibronectin, CD11c, IDO1, CD27) as predictor for survival in larger cancer datasets. The closest that comes to mind is the Lung Adenocarcinoma from TCGA, which has vast survival, transcriptomics and proteomics data, and it histologically very similar to NPC. However, the authors may indeed choose a dataset they believe to be more appropriate for the survival analysis based on these markers.

- Figure 1B and 2B: protein names are overlapping and written in faint color, so are impossible to read. The authors should use some sort of repositioning of the tables. Since the authors seem to be using R for this plot, a suggestion could be using the textplot() function from the wordcloud package, or the ggrepel package, or even manually separating the overlapping labels.

- Line 211: unless there are different numbers for variables, it is not necessary to specify two thresholds for Pearson's correlation significance. The authors can simply specify that significance was defined as |r|>0.7, which corresponds to p-value=0.xx (to be calculated). Also, please specify the absolute value around the coefficient |r|, otherwise it looks like only positive correlations were investigated.

Author Response

Dear reviewer,

Kind regards,

Authors

Round 2

Reviewer 2 Report

No other questions. 

Author Response

Dear reviewer, 

Thank you for your comments and feedback. 

Best regards,

Authors

Reviewer 3 Report

The authors have made some changes to the text, but somehow they have not answered to my comments with actual changes to the manuscript.

- For example, I still see overlapping labels in figure 1b and 2b, which the authors claim to have fixed.

- Also, I would not limit the survival analysis performed by the authors on the TCGA cohorts to the response to reviewers, since the same doubts could very well be raised by readers in the future. The survival analysis provided as Figures C and D in the authors' response, together with the corresponding text and discussion, should be cited in the main text and added to the supplementary materials of the manuscript.

- I still see plenty of references to Epstein Barr Virus in the abstract, while the paper does not discuss its role in nasopharyngeal spatial development or etiology, and the virus has no clear causal role or scientific implications in the presented analysis.

- Also, as I wrote in the previous round of reviews the authors must specify which exact p-value corresponds to |r|>0.7, not only a generic "p<0.05". For example, for r=0.7, at n=10 p=0.02420634; at n=100 p=5.329041e-16; at n=1000 p=4.307596e-148. The conversion of correlation r, given n (number of samples), to a p-value can be obtained for example with the R function corto::r2p()

Author Response

Dear reviewer,

Best regards,

Authors

Round 3

Reviewer 3 Report

The authors have amended their manuscript in the latest review, thereby providing an improved version of their paper.